# Taxed and untaxed beverage intake by South African young adults after a national sugar-sweetened beverage tax: A before-and-after study

Michael Essman[1,2], Lindsey Smith Taillie[1,2], Tamryn Frank[3], Shu Wen Ng[1,2], Barry M. Popkin[1,2], Elizabeth C. Swart[3]*

1 Department of Nutrition, Gillings School of Global Public Health, University of North Carolina at Chapel Hill, Chapel Hill, North Carolina, United States of America, 2 Carolina Population Center, University of North Carolina at Chapel Hill, Chapel Hill, North Carolina, United States of America, 3 Faculty of Community and Health Sciences, University of the Western Cape, Bellville, South Africa

* rswart@uwc.ac.za

## Abstract

### Background

In an effort to prevent and reduce the prevalence rate of people with obesity and diabetes, South Africa implemented a sugar-content-based tax called the Health Promotion Levy in April 2018, one of the first sugar-sweetened beverage (SSB) taxes to be based on each gram of sugar (beyond 4 g/100 ml). This before-and-after study estimated changes in taxed and untaxed beverage intake 1 year after the tax, examining separately, to our knowledge for the first time, the role of reformulation distinct from behavioral changes in SSB intake.

### Methods and findings

We collected single-day 24-hour dietary recalls from repeat cross-sectional surveys of adults aged 18–39 years in Langa, South Africa. Participants were recruited in February–March 2018 (pre-tax, $n = 2,459$) and February–March 2019 (post-tax, $n = 2,489$) using door-to-door sampling. We developed time-specific food composition tables (FCTs) for South African beverages before and after the tax, linked with the diet recalls. By linking pre-tax FCTs only to dietary intake data collected in the pre-tax and post-tax periods, we calculated changes in beverage intake due to behavioral change, assuming no reformulation. Next, we repeated the analysis using an updated FCT in the post-tax period to capture the marginal effect of reformulation. We estimated beverage intake using a 2-part model that takes into consideration the biases in using ordinary least squares or other continuous variable approaches with many individuals with zero intake. First, a probit model was used to estimate the probability of consuming the specific beverage category. Then, conditional on a positive outcome, a generalized linear model with a log-link was used to estimate the continuous amount of beverage consumed. Among taxed beverages, sugar intake decreased significantly ($p < 0.0001$) from 28.8 g/capita/day (95% CI 27.3–30.4) pre-tax to 19.8 (95% CI

**Data Availability Statement:** Data analyzed for this paper form part of a primary project which is currently being written up in other publications.

Data used for this paper will therefore be available upon request and granted for replication purposes. Data are available from the UNC Carolina Digital Repository (https://cdr.lib.unc.edu/). For data inquiries, please contact Donna Miles (drmiles@email.unc.edu) and Jessica Ostrowski (jessica.ostrowski@unc.edu).

**Funding:** Funding for this study comes from Bloomberg Philanthropies (https://www.bloomberg.org/; received by BP), including Subcontract #5108311 with the University of the Western Cape. This research also received support from the Population Research Infrastructure Program awarded to the Carolina Population Center (P2C HD050924) at The University of North Carolina at Chapel Hill by the Eunice Kennedy Shriver National Institute of Child Health and Human Development. ME was supported by a Predoctoral Fellowship from NIH Training Grant T32 HL129969. Other support includes student scholarships from International Development Research Center (IDRC) Project number 108425-001 and the DST/NRF Centre of Excellence in Food Security project number 180401 (https://www.uwc.ac.za/Faculties/CHS/soph/Pages/The%20Centre-of-Excellence-in-Food-Security.aspx). The funders had no role in study design, data collection and analysis, decision to publish, or preparation of the manuscript.

**Competing interests:** We have read and understood PLOS Medicine's policy on declaration of interests and ME, LST, TF, SWN, and ECS declare that they have no competing interests. BP is on the PLOS Medicine editorial board and otherwise has no competing interests.

**Abbreviations:** FCT, food composition table; HPL, Health Promotion Levy; LSM, Living Standards Measure; NFP, nutrition facts panel; SSB, sugar-sweetened beverage.

18.5–21.1) post-tax. Energy intake decreased ($p < 0.0001$) from 121 kcal/capita/day (95% CI 114–127) pre-tax to 82 (95% CI 76–87) post-tax. Volume intake decreased ($p < 0.0001$) from 315 ml/capita/day (95% CI 297–332) pre-tax to 198 (95% CI 185–211) post-tax. Among untaxed beverages, sugar intake increased ($p < 0.0001$) by 5.3 g/capita/day (95% CI 3.7 to 6.9), and energy intake increased ($p < 0.0001$) by 29 kcal/capita/day (95% CI 19 to 39). Among total beverages, sugar intake decreased significantly ($p = 0.004$) by 3.7 (95% CI −6.2 to −1.2) g/capita/day. Behavioral change accounted for reductions of 24% in energy, 22% in sugar, and 23% in volume, while reformulation accounted for additional reductions of 8% in energy, 9% in sugar, and 14% in volume from taxed beverages. The key limitations of this study are an inability to make causal claims due to repeat cross-sectional data collection, and that the magnitude of reduction in taxed beverage intake may not be generalizable to higher income populations.

## Conclusions

Using a large sample of a high-consuming, low-income population, we found large reductions in taxed beverage intake, separating the components of behavioral change from reformulation. This reduction was partially compensated by an increase in sugar and energy from untaxed beverages. Because policies such as taxes can incentivize reformulation, our use of an up-to-date FCT that reflects a rapidly changing food supply is novel and important for evaluating policy effects on intake.

## Author summary

### Why was this study done?

- In 2018, South Africa became the first sub-Saharan African country to implement a sugary beverage tax to reduce consumption of added sugars.

- Sugar-sweetened beverage (SSB) taxes have been shown to reduce purchases, but few studies have included dietary intake, and they are focused in higher income countries like the United States. To our knowledge, this is the first study to examine the effects of an SSB tax using detailed 24-hour dietary recall data focused on a low-income population from the Global South.

- To our knowledge, this is the first study to evaluate real-world changes from an SSB tax policy by empirically quantifying how much of the overall change in sugar intake from taxed beverages came from consumers' behavioral changes versus reformulation of beverages.

### What did the researchers do and find?

- Using dietary intake data collected from a low-income South African township before and after the SSB tax was implemented, we examined changes in sugar, calorie, and volume intake from taxed and untaxed beverages.

- We used detailed brand-product-specific nutrition facts panel data collected at baseline and a year after tax implementation to allow us to examine both behavioral change and reformulation.

- We found the intake of sugar, calories, and volume of taxed beverages decreased by 9.0 g (31%), 39 kcal (33%), and 117 ml (37%) per capita per day, respectively. The intake of sugar, calories, and volume of untaxed beverages increased by 5.3 g (36%), 30 kcal (29%), and 339 ml (58%) per capita per day, respectively. Water accounted for the majority (52%) of the pre–post difference in volume of untaxed beverage intake.

- Of the 9.0 g per capita per day (31%) total reduction in sugar from taxed beverages, 6.4 g per capita per day (22%) was due to behavioral differences. When accounting for reformulation, there was an additional 2.6 g per capita per day (9%) reduction in the post-tax period.

### What do these findings mean?

- SSB taxes were associated with reduced sugary drink intake in a low-income population within a middle-income country.

- We quantified the observed changes in beverage intake due to reformulation and behavior change, thereby examining the 2 primary goals of an SSB tax based on sugar concentration.

- Future research will be needed to understand how responses to sugar-based beverage taxes may vary by socioeconomic status.

## Introduction

Consumption of sugar-sweetened beverages (SSBs) is increasing particularly rapidly in low- and middle-income countries [1]. Given the major impact of SSBs on obesity and many key noncommunicable diseases, and building upon evidence that national SSB taxes reduced SSB purchases in Mexico and Chile in 2014, over 40 countries have now implemented a national SSB tax or increased an existing tax in the last decade [2–9]. South Africa, with one of the highest SSB consumption rates in Africa and a growing prevalence of type 2 diabetes, is the first sub-Saharan African country to institute a sugary beverage tax [10], implemented in April 2018.

The South African tax, called the Health Promotion Levy (HPL), is one of the first SSB taxes to be based on sugar content, applying a fixed 2.1-cent tax rate for every gram of sugar (both intrinsic and added) above a 4 g/100 ml threshold [11]. Early calculations suggest that the average tax rate is approximately 10%. A similar threshold-based multi-tiered SSB levy has been passed in the United Kingdom, but the HPL structure is potentially stronger as each additional gram per 100 ml imposes a greater tax [12]. Taxes based on sugar concentration may lead to greater impact on health outcomes than volume-based taxes, both by reducing the purchases of SSBs and by encouraging product reformulation to lessen the tax burden, thereby reducing excessive added sugars [13,14].

Heretofore, no major national tax evaluation has examined changes in dietary intake of taxed beverages as a key outcome. This study is unique in using detailed dietary intake data on

a sample of low-income young adults in South Africa. This is important because two-thirds of cardiovascular deaths occur in middle- and high-income countries, and within those countries it is the lowest income communities with the highest risk [15]. To our knowledge, this study is also the first SSB tax evaluation of a national tax to use detailed 24-hour dietary recall data. Previous work estimated crude store intercepts and used SSB frequency questionnaires for small city evaluations, but these methods miss other sources of consumption and do not represent key segments of a country [16–20]. Other studies have used high-quality household purchase data or cruder aggregate sales data [6,7,21,22]. While these data are important, they exclude many sources of SSBs that dietary measures overcome. For example, collecting purchase information through exit interviews at stores could miss other locations where participants shopped. Food purchase studies can also miss incidental beverage purchases and underestimate the total consumption. Another source of consumption data, aggregate sales from Euromonitor International, can miss the sales from vending and smaller nonchain stores. In contrast, individual dietary intake data record what each individual actually consumes, regardless of where it was purchased. Although 24-hour recall data are susceptible to both random and systematic errors, they are considered by the National Cancer Institute the least biased self-report dietary intake instruments and the most appropriate for evaluating the effects of an intervention [23].

The objectives of our study are to estimate differences in total sugar, energy, and volume from taxed (>4 g sugar/100 ml ready-to-drink), untaxed (≤4 g sugar/100 ml ready-to-drink), and total beverages using 24-hour dietary recall data from the Langa township of South Africa before and 1 year after the HPL. A novel contribution of our study to the SSB tax literature involves linking updated food composition tables (FCTs) with dietary recall data. We developed time-specific FCTs for South African beverages before and after the HPL, which are linked with the dietary assessment instruments. This allows us to examine separately for the first time the role of reformulation as distinct from behavioral changes in SSB intake.

## Methods

This study was part of a student dissertation and not registered in a public repository. However, the original proposal of the study design and analytical plan from the dissertation proposal can be found in S1 Protocol. This study is reported as per the Strengthening the Reporting of Observational Studies in Epidemiology (STROBE) guideline (S1 STROBE Checklist).

### Data sources and measures

To evaluate the HPL, we analyzed single-day 24-hour dietary recalls from repeat cross-sectional surveys of young township adults aged 18–39 years living in the lower income Langa township near Cape Town, South Africa (S1 Survey). Our study population in the Langa township was selected due to the stability of the community for repeated data collections across time and because it contains a large number of young adults who are heavy consumers of SSBs. At last count, Langa had 17,402 households and 52,401 inhabitants (50.4% female), of whom 99.1% were of Black African race [24].

Data were collected in February–March 2018 (pre-tax, 2 months before the SSB tax implementation; $n = 2,481$) and a post-tax survey 12 months later in February–March 2019 ($n = 2,507$) to measure differences in beverage consumption following the HPL (Fig 1). Twenty-two diet records in the pre-tax group (0.9%) and 18 diet records in the post-tax group (0.7%) were dropped for reporting less than 400 daily kcal. Thus, 2,459 and 2,489 diet recalls were included in the final analysis for pre-tax and post-tax, respectively (Table 1). Because the

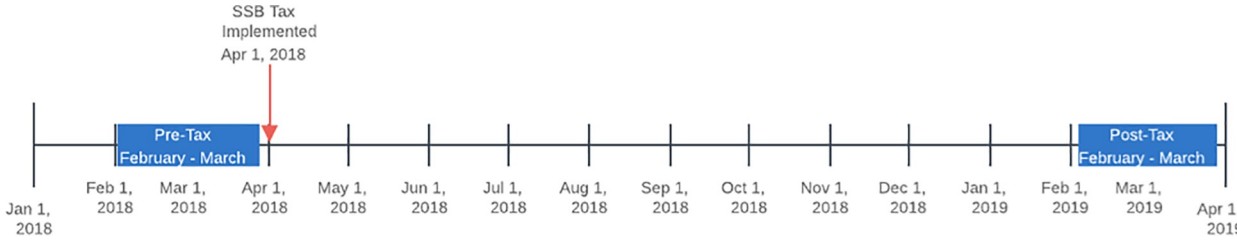

**Fig 1. Timeline for data collection.** SSB, sugar-sweetened beverage.

survey was designed to capture young adults' SSB intake, the only eligibility criterion was being 18–39 years of age. Participants were recruited using a door-to-door sampling method of all identifiable households in Langa until the target sample size of approximately 2,500 households was achieved at each collection period. Fieldworkers approached every household in Langa to ascertain if a household member met the entry requirement (aged 18–39 years) and if that individual was willing to participate. Only 1 diet assessment was completed for 1 individual within a given household. Where 2 qualifying participants were present in the household, the first qualifying participant was selected if the household number in the survey was an uneven number, and the second participant was selected if the household number was an even number. If 3 or more qualifying participants were present in the household, a random numbers list was used to select the respondent. The household questionnaire was conducted digitally and included a geolocation. The geolocations provided us with maps to ensure that we covered all areas. Participants received a supermarket voucher worth R30 (US$2.19) after participating. At the post-tax data collection, participants were asked whether they had been previously surveyed. However, the data are not longitudinal because individuals' diet surveys cannot be linked across time.

To record anthropometry, fieldworkers used standardized scales and stadiometers to record the weight and height of each participant after the diet recall was completed, measuring each twice. Body mass index (BMI) was calculated by dividing weight in kilograms by the square of height in meters, using the average of the 2 measurements.

**Table 1. Sociodemographic information for Langa survey of adults aged 18–39 years.**

| Variable | Pre-tax (n = 2,459) | Post-tax (n = 2,489) |
|---|---|---|
| Male | 34.8% | 34.8% |
| Female | 65.2% | 65.2% |
| LSM category[1] | | |
| LSM 3 | 1.2% | 1.6% |
| LSM 4 | 13.1% | 19.1%* |
| LSM 5 | 34.4% | 49.7%* |
| LSM 6 | 34.9% | 27.2%* |
| Missing/incomplete data | 16.4% | 2.5 |
| Age (years) | 27.9 (6.0) | 27.8 (6.2) |
| Male BMI (kg/m$^2$) | 22.8 (4.4) | 22.6 (4.1) |
| Female BMI (kg/m$^2$) | 29.6 (7.3) | 30.3 (7.5) |

Data given as percent or mean (SD).

[1]South African Living Standards Measure (LSM) [30].

*Indicates statistically significant difference ($p < 0.001$) from pre-tax using Fisher exact test.

**Measuring dietary intake.** For the diet assessment, 24-hour diet recalls were conducted by interviewers with nutrition training. Participants reported what foods and drinks were eaten, how foods and beverages were prepared, whether anything was added, and the quantity consumed. A multiple-pass approach, including detailed prompting, was used to enhance completeness.

**Linking FCTs to beverage categories.** We created composite nutritional records for beverages based on the current food supply and consumer purchases. First, nutrition facts panel (NFP) data were collected from South African grocery stores in February and March 2018. This was repeated exactly a year later in February and March 2019. The stores were sampled in 2 ways. First, we obtained permission and collected all NFP data from beverages at all major chains functioning in the Cape Town region. Second, we visited small stores in the sample township area of Langa to ensure we collected any beverages missed in the chain stores that may have been consumed by our sample population. Products from each round of NFP data collection were linked to a database of 114 beverage codes for creating a South Africa FCT. Fieldworkers who coded the 24-hour recalls created codes by brand name for each SSB, allowing linkages between NFP data and dietary intake data. Fieldworkers always prompted whether beverages were diet or regular beverage types to ensure reported beverages were correctly placed in the beverage code that was above or below the 4 g/100 ml taxation threshold. Each beverage code was given an average nutrient profile, weighted by household purchase data from Kantar Worldpanel, a panel dataset of household packaged food and beverage purchases. The pre-tax beverage FCT was linked to 2018 NFP–Kantar data, and the updated post-tax beverage FCT was linked to 2019 NFP–Kantar data. Each FCT beverage code was also categorized by taxation status. Beverage taxation status was determined by a 2-step process: (1) whether the product category is taxable, as 100% fruit juice and unsweetened milks are exempt from the HPL, and (2) among all other beverages that are taxable, those with a total sugar concentration greater than 4 g/100 ml are classified as taxed and those with 4 g/100 ml or less are untaxed. We considered all tax-exempt beverages and beverages with total sugar $\leq$ 4 g/100 ml as untaxed. Beverage categories were ultimately analyzed as either taxed or untaxed according to the beverage grouping system (S1 Table).

## Analytical approach

All analyses were conducted in Stata, version 16 [25]. Our key outcome variables include intake of total sugar (grams), energy (kilocalories), and volume (milliliters) for total beverages, taxed beverages, untaxed beverages, and subcategories of taxed and untaxed beverages. Intake estimates for these beverage categories were made for the pre-tax and post-tax collection periods. There were few deviations from the initial analysis plan outlined in S1 Protocol. For this study, we added total beverage intake as an outcome to demonstrate the total effect of the policy on beverage intake, not just taxed and untaxed beverage intake alone. We do not report results for beverage subcategories, to maintain focus and clarity on changes in taxed, untaxed, and total beverage intake after the HPL.

**Estimation of beverage intake.** We estimated beverage intake from 24-hour recalls using a 2-part model implemented with the Stata twopm command to account for beverage groups that have a high percentage of non-consumers, meaning study participants who did not report a particular beverage category consumed [26]. We used a probit model for the first part (likelihood of consumption), and conditional on consumption, we used a generalized linear model with log-link, which gives unbiased estimates of amount consumed, for the second part [27]. The gamma distribution with a log-link is appropriate for data that are continuous, positive, and right-skewed, and these assumptions fit our intake data. Primary outcomes are reported

with 95% confidence intervals, and statistically significant differences between groups were calculated using a Wald test [28]. The main comparisons were whether the predicted mean intakes of sugar, energy, and volume were different in the post-tax period compared to the pre-tax period. Models were adjusted for age (continuous, range 18–39), sex, weekday versus weekend of intake (binary), average daily temperature (obtained from National Centers for Environmental Information [29]), and socioeconomic status using the Living Standards Measure (LSM) of the South African Audience Research Foundation (previously the South African Advertising Research Foundation) [30]. LSM is a standardized composite measure commonly used to segment based on socioeconomic status in South Africa and has been widely used by industry since 1989 [31]. It is based on 29 items, including mostly household assets [30]. Other measures such as income are considered less reliable in South Africa for categorizing socioeconomic status. With a total range from 1 to 10, the range of LSM in our study, from 3 to 6, includes low to middle socioeconomic status in South Africa. LSM 3 and LSM 4 were combined to increase power for comparisons with the lowest group. Models are adjusted for the same covariates in both steps. We did not adjust for intake from foods as we were interested in capturing the effects of the policy on beverage intake, which could include substitution effects.

In our main analyses, pre–post beverage intake comparisons were made using the pre-tax beverage FCT linked to both the pre-tax beverage intake data and the post-tax beverage intake data. Thus, any changes in beverage intake would be due to behavioral change alone, since we effectively assume no reformulation. Next, we analyzed post-tax beverage intake linked to the updated beverage FCT to reflect the nutritional composition of beverages at each time point. This analysis reflects the combined effects of reformulation and behavioral change.

**Beverage intake by socioeconomic status.** To determine whether our results differed by socioeconomic category (LSM), an interaction term between time of data collection and LSM category was added to adjusted models, and tested for statistical significance using a Wald test.

**Sensitivity analyses.** We conducted a series of sensitivity analyses to test modeling decisions. We tested whether the 2-part model was less appropriate for modeling total beverage intake, with a percentage of consumers over 90%, comparing results from the 2-part model to a generalized linear model with a log-link alone. We investigated adding BMI as a covariate in our models, in case there were differences in reporting of beverage intake by body mass. We also excluded participants who were present both at pre-tax and post-tax data collection (12.4% of post-tax sample), in case participants who repeated the survey responded differently from those taking it for the first time. A part of our sample data in the post-tax period was collected when a drought was in function and strict water controls were enforced. We tested whether our results were confounded by water shortages by controlling for a variable indicating whether participants changed their beverage consumption due to drought. Finally, to test for the absence of bias due to missing LSM data, we tested whether missingness on LSM depended on the outcome (beverage intake), conditional on the covariates.

## Ethical approval

The study protocol, questionnaires, procedures, and informed consent forms were approved by the Biomedical Research Ethics Committee of the University of the Western Cape (#BM17/8/20 and #BM18/6/2) as well as by University of North Carolina at Chapel Hill Institutional Review Board (#18–2028). All participants gave written informed consent to enroll in the study.

## Patient and public involvement

Participants and the public were not involved in the design, conduct, reporting, or dissemination plans of our research.

## Results

Study population characteristics are presented in Table 1. From pre-tax to post-tax, the percent of respondents in LSM 4 and LSM 5 increased ($p < 0.001$), and the percent of respondents in the highest category, LSM 6, decreased ($p < 0.001$). There were no significant differences between the 2 time points for any other sociodemographic characteristics.

### Adjusted results

**Total effects for sugar, energy, and volume intakes for taxed beverages.** Sugar intake from taxed beverages decreased ($p < 0.0001$) from 28.8 g/capita/day (95% CI 27.3 to 30.4) pre-tax to 19.8 (95% CI 18.5 to 21.1) post-tax, a 31.4% reduction (Table 2). Energy intake from taxed beverages decreased ($p < 0.0001$) from 121 kcal/capita/day (95% CI 114 to 127) pre-tax to 82 (95% CI 76 to 87) post-tax, a 32.5% reduction (Table 2). Volume intake from taxed beverages decreased ($p < 0.0001$) from 315 ml/capita/day (95% CI 297 to 332) pre-tax to 198 (95% CI 185 to 211) post-tax, a 37.1% reduction (Table 2). Confidence intervals for absolute differences are reported in Table 2. Sugar, energy, and volume intakes of taxed beverage subcategories are reported in S2–S4 Tables.

**Total effects for sugar, energy, and volume intakes for untaxed beverages.** Sugar intake from untaxed beverages increased ($p < 0.0001$) from 15.0 g/capita/day (95% CI 13.9 to 16.0) pre-tax to 20.3 (95% CI 18.2 to 21.4) post-tax, a 35.5% increase. Energy intake from untaxed beverages increased ($p < 0.0001$) from 105 kcal/capita/day (95% CI 99 to 112) pre-tax to 135 (95% CI 128 to 141) post-tax, a 28.6% increase. Volume intake from untaxed beverages increased ($p < 0.0001$) from 587 ml/capita/day (95% CI 563 to 610) pre-tax to 926 (95% CI 899 to 953) post-tax (Table 2). Water accounted for the majority (52%) of this pre–post

**Table 2. Model adjusted predicted values for sugar, energy, and volume intakes for taxed, untaxed, and total beverages.**

| Category | Sugar intake, g/capita/day (95% CI) | Energy intake, kcal/capita/day (95% CI) | Volume intake, ml/capita/day (95% CI) |
|---|---|---|---|
| **Taxed beverages** | | | |
| Pre-tax | 28.8 (27.3 to 30.4) | 121 (114 to 127) | 315 (297 to 332) |
| Post-tax | 19.8 (18.5 to 21.1) | 82 (76 to 87) | 198 (185 to 211) |
| Absolute difference | −9.1 (−11.2 to −6.9)** | −39 (−48 to −30)** | −117 (−139 to −94)** |
| **Untaxed beverages** | | | |
| Pre-tax | 15.0 (13.9 to 16.0) | 105 (99 to 112) | 587 (563 to 610) |
| Post-tax | 20.3 (19.2 to 21.4) | 135 (128 to 141) | 926 (899 to 953) |
| Absolute difference | 5.3 (3.7 to 6.9)** | 29 (19 to 39)** | 340 (303 to 376)** |
| **Total beverages** | | | |
| Pre-tax | 43.8 (41.9 to 45.7) | 226 (217 to 235) | 901 (876 to 927) |
| Post-tax | 40.1 (38.5 to 41.6) | 216 (208 to 224) | 1,124 (1097 to 1151) |
| Absolute difference | −3.7 (−6.2 to −1.2)* $p = 0.004$ | −10 (−23 to 2) | 223 (184 to 261)** |

From models adjusting for age, sex, weekday versus weekend, average daily temperature, and socioeconomic status (Living Standards Measure). Exact $p$-values are reported unless $p < 0.001$.

*Statistically significant difference at the $p < 0.01$ level.

**Statistically significant difference at the $p < 0.0001$ level.

difference in untaxed beverage intake (S4 Table). Sugar, energy, and volume intakes of untaxed beverage subcategories are reported in S2–S4 Tables.

**Total effects for sugar, energy, and volume intakes for total beverages.** Sugar intake from total beverages significantly decreased ($p = 0.004$) from 43.8 g/capita/day (95% CI 41.9 to 45.7) pre-tax to 40.1 (95% CI 38.5 to 41.6) post-tax, a 8.4% reduction (Table 2). However, there was no significant change in energy intake comparing pre-tax (226 kcal/capita/day; 95% CI 217 to 235) with post-tax (216 kcal/capita/day; 95% CI 208 to 224; $p = 0.1$) (Table 2). Volume intake was 223 ml/capita/day (95% CI 184 to 261) greater post-tax, partially driven by the large increase in water intake (S4 Table).

**Reformulation effects for sugar, energy, and volume intakes.** For taxed beverages, we found greater pre–post differences when accounting for reformulation. Sugar intake from beverages was 28.8 g/capita/day (95% CI 27.3 to 30.4) in the pre-tax period and 22.4 (95% CI 21.1 to 23.8) in the post-tax period (−22.2%). When accounting for reformulation, there was a 9.2% additional reduction compared to the pre-tax period, to 19.8 g/capita/day (95% CI 18.5 to 21.1) in the post-tax period (Fig 2). Energy intake was 121 kcal/capita/day (95% CI 114 to 127) in the pre-tax period and 92 (95% CI 86 to 97) in the post-tax period (−24.1%). When accounting for reformulation, there was an 8.4% additional reduction compared to the pre-tax period, to 82 kcal/capita/day (95% CI 76 to 87) in the post-tax period (Fig 3). Volume intake was 315 ml/capita/day (95% CI 297 to 352) in the pre-tax period and 241 (95% CI 227 to 256) in the post-tax period (−23.3%). When accounting for reformulation, there was a 13.7% additional reduction compared to the pre-tax period, to 198 ml/capita/day (95% CI 185 to 211) in the post-tax period (Fig 4).

For untaxed beverages, energy intake was 105 kcal/capita/day (95% CI 99 to 112) in the pre-tax period and 141 (95% CI 134 to 148) in the post-tax period (+33.9%). When accounting for reformulation, there was less of an increase, to 135 (+27.5%) kcal/capita/day (95% CI 76 to 87) in the post-tax period (Fig 3).

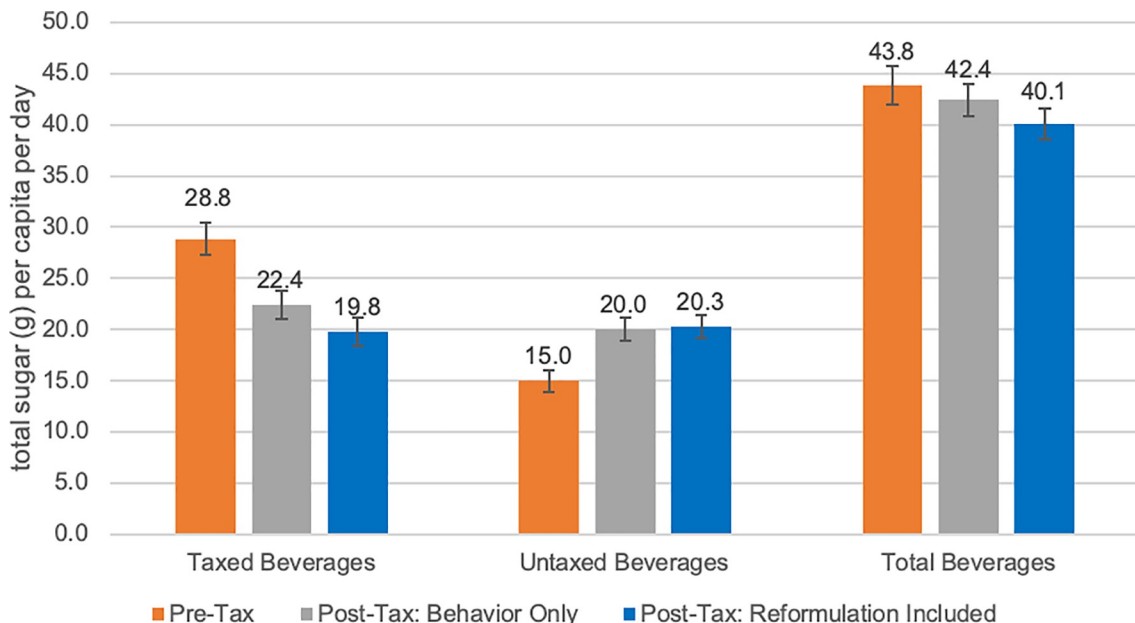

**Fig 2. Estimated daily intake in grams of sugar per capita from taxed, untaxed, and total beverages.** "Post-Tax: Behavior Only" refers to estimates of beverage intake using only the pre-tax food composition table (FCT), which keeps nutrient content constant. "Post-Tax: Reformulation Included" refers to estimates of beverage intake using the post-tax FCT, which accounts for reformulation changes in the nutrient content of beverages. Error bars represent 95% confidence intervals.

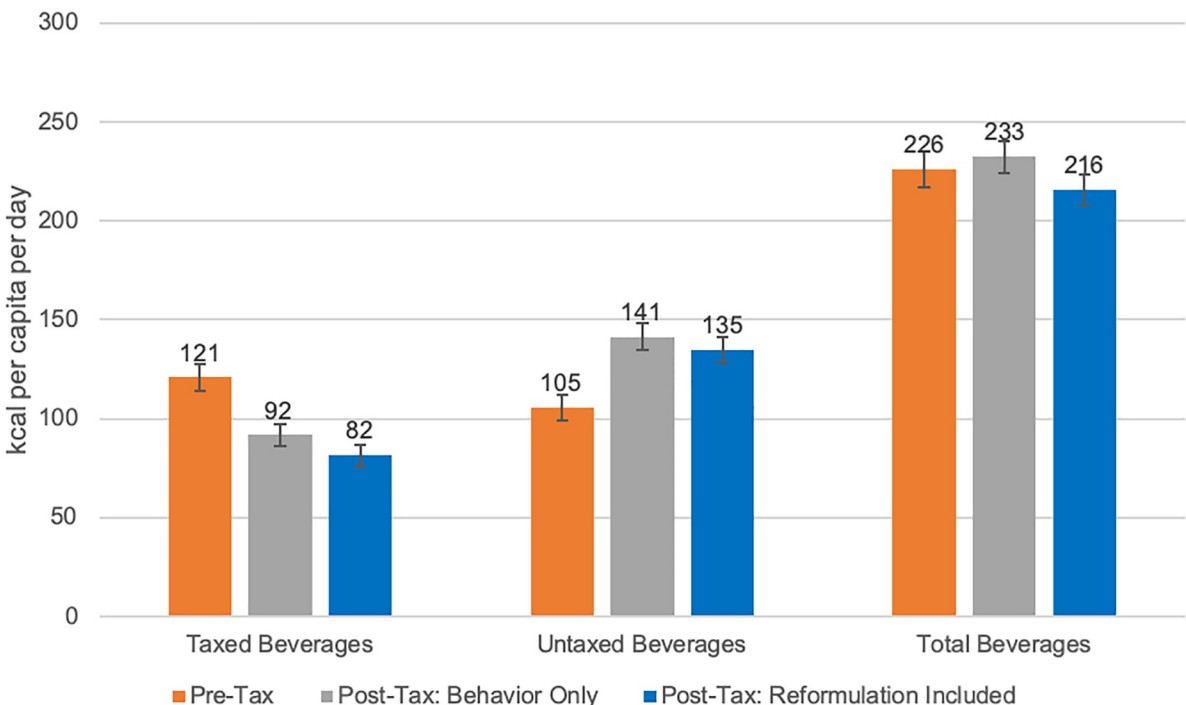

**Fig 3. Estimated daily intake in energy (kilocalories) per capita from taxed, untaxed, and total beverages.** "Post-Tax: Behavior Only" refers to estimates of beverage intake using only the pre-tax food composition table (FCT), which keeps nutrient content constant. "Post-Tax: Reformulation Included" refers to estimates of beverage intake using the post-tax FCT, which accounts for reformulation changes in the nutrient content of beverages. Error bars represent 95% confidence intervals.

For total beverages, sugar intake was 43.8 g/capita/day (95% CI 41.9 to 45.7) in the pre-tax period and 42.4 (95% CI 40.9 to 44.0) in the post-tax period (−3.2%). When accounting for reformulation, there was a 5.4% additional reduction to 40.1 g/capita/day (95% CI 38.5 to 41.6) in the post-tax period, making the total difference from pre-tax statistically significant (*p* = 0.004) (Fig 2). Energy from total beverages was 226 kcal/capita/day (95% CI 217 to 235) in the pre-tax period and 233 (95% CI 224 to 241) in the post-tax period. When accounting for reformulation, there was a 4.4% total reduction to 216 kcal/capita/day (95% CI 208 to 224) in the post-tax period, although this reduction was not statistically significant (Fig 3).

**Beverage intake by LSM.**   All socioeconomic groups consumed significantly less sugar, energy, and volume from taxed beverages post-tax compared to pre-tax (S5 Table). For untaxed beverages, all LSM groups significantly increased their intake of sugar, energy, and volume post-tax compared to pre-tax (S5 Table). For total beverages, only LSM 6 had a statistically significant reduction in sugar intake post-tax (S5 Table). There were no differences in the magnitude of change between LSM groups for taxed, untaxed, or total beverage intakes.

**Sensitivity analyses.**   We tested whether the 2-part model was less appropriate for modeling total beverage intake, with a percentage of consumers over 90%. We found a less than 1-kcal difference between results using the 2-part model and solely a generalized linear model with a log-link. The sensitivity analysis including BMI in the beverage intake model estimated an additional reduction of 0.5 g of sugar (2 kcal/capita/day) post-tax compared to pre-tax for taxed beverages. Therefore, to be more conservative in our conclusions, we did not include BMI in the model for our main results. We also tested whether our results were affected by excluding participants who were present at both pre-tax and post-tax data collection (12.4%) and estimated an additional 2 kcal/capita/day reduction when excluding any repeats.

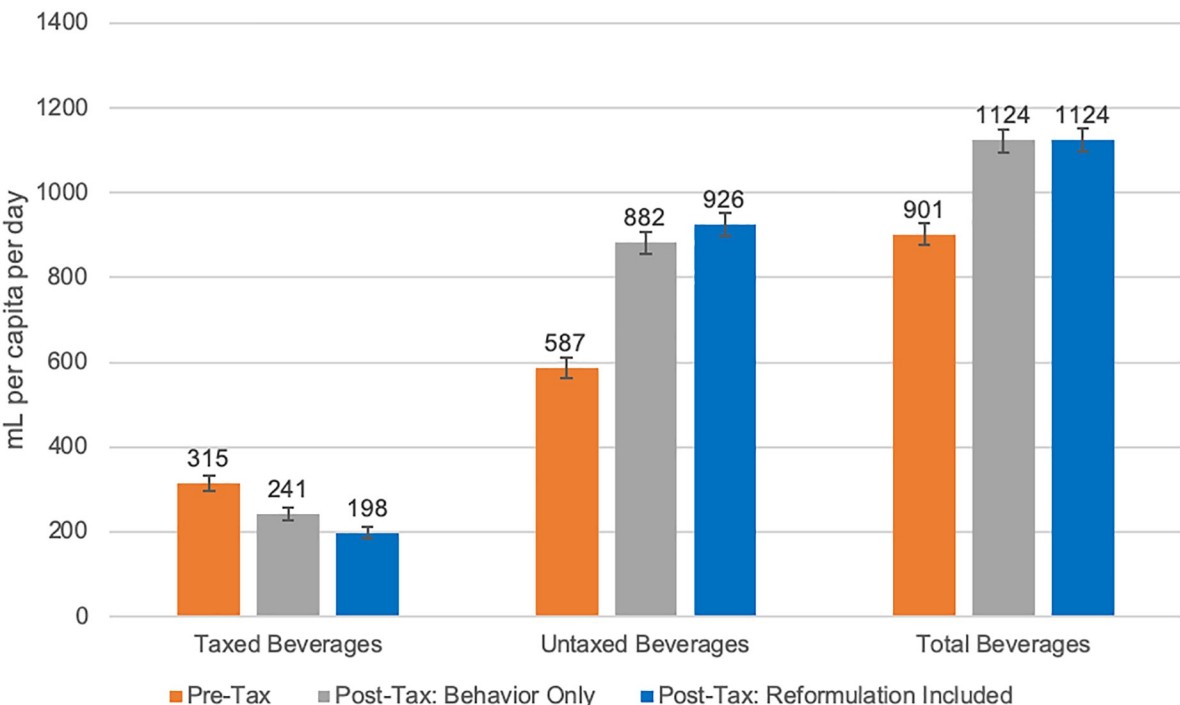

**Fig 4. Estimated daily intake in milliliters per capita from taxed, untaxed, and total beverages.** "Post-Tax: Behavior Only" refers to estimates of beverage intake using only the pre-tax food composition table (FCT), which keeps nutrient content constant. "Post-Tax: Reformulation Included" refers to estimates of beverage intake using the post-tax FCT, which accounts for reformulation changes in the nutrient content of beverages. Error bars represent 95% confidence intervals.

Therefore, to be more conservative in our conclusions, we kept the entire sample. When we controlled for whether participants changed their beverage consumption due to drought, the results shifted by less than 1 kcal when estimating energy intake from taxed beverages. We found missingness on LSM did not depend on the outcome (beverage intake). This suggests our intake estimates are not biased due to the missing LSM data.

## Discussion

This study evaluated the effects of the South African HPL on taxed, untaxed, and total beverage intake by young adults in a Cape Town township 1 year after the tax was implemented. Our novel contribution to the SSB tax literature is the use of updated FCTs linked to dietary intake data to more closely track changes in the food supply after the tax, thus allowing us to separate the effects of changing consumer behavior and industry reformulation. We found that the HPL was followed by a 9.1 g/capita/day (31.4%) reduction in sugar intake, a 39 kcal/capita/day (32.5%) reduction in energy intake, and a 117 ml/capita per day (37.1%) reduction in volume intake from taxed beverages at 1 year post-implementation. We found a 5.3 g/capita/day increase in sugar intake and a 29 kcal/capita/day increase in energy intake from untaxed beverages that partially compensated for this large reduction in taxed beverage intake. Across all beverages, we found a total reduction of 3.7 g/capita/day (8.4%) in sugar consumption ($p$ = 0.004). Our results indicate overall that the behavior change of these adults was associated with reductions of 22% in taxed beverage sugar intake and 3.2% in total beverage sugar intake compared to pre-tax levels, and our estimate of reformulation was the remainder of the post-tax difference.

To our knowledge, this is the first detailed dietary intake survey to find significant results of a national tax effort on dietary intake [32]. The present study collected 24-hour recalls, which are more suitable for estimating mean intakes in a population than frequency questionnaires, and had a larger sample size than earlier studies [33]. For example, Silver et al. found a 19.8% reduction in volume intake and a 13.3% reduction in caloric intake of SSBs following the SSB tax in Berkeley, CA, US, but these findings lacked precision and did not reach statistical significance, potentially due to low baseline intakes in Berkeley (45 kcal/capita/day) compared to our larger high-consuming population pre-tax (121 kcal/capita/day) [21]. Our study population was also a low-income community, and greater reductions in taxed beverage intake could be due to greater price sensitivity. Results from Mexico have also shown the greatest reductions in SSB purchases among the lowest income groups following a tax [34].

Our key methodological contribution is the ability to separate behavioral change from reformulation effects using time-specific FCTs linked to each dietary intake collection period. Accurate FCTs are necessary to calculate nutrient intakes and total energy from 24-hour recalls and frequency questionnaires [35,36]. A study on the threshold-based SSB tax in the United Kingdom noted the combined effects of behavioral change and reformulation in reducing sugar consumption from beverages [37], but to our knowledge, this is the first study to separately quantify the contribution of each in the context of a real-world policy evaluation of dietary intake. Behavioral change accounted for reductions of 24% in energy, 22% in sugar, and 23% in volume compared to pre-tax, while reformulation accounted for additional reductions of 8% in energy, 9% in sugar, and 14% in volume from taxed beverages. Changes in volume intake due to reformulation should be interpreted as effects of updating sugar concentration that led to changed taxed beverage classification. For example, a beverage from the pre-tax period could become categorized as untaxed if it was reformulated below the 4 g/100 ml threshold.

It is important to note that despite the utility of these new methods, we may still be underestimating the effects of the HPL on taxed beverage intake in our study sample. Recent analyses of the impact of the UK soft drink levy found significant reductions in sugar levels in SSBs between the announcement and implementation of the levy [37,38]. The initial announcement of a plan to tax SSBs was made by Finance Minister Pravin Gordhan during a budget speech in February 2016, over 2 years before the SSB tax was ultimately implemented. Our study did not capture any preemptive reductions in beverage sugar content such as have been suggested in other countries [37,38]. Other data sources such as purchase data will be important for capturing changes in the nutrition content of beverages in the time period between the announcement and implementation of the tax.

Other work used a different decomposition method to capture reformulation of the food supply, although for purchase data [39,40]. Griffith and colleagues used a decomposition method that was useful for measuring changes in the salt density of food purchases [39]. Both Griffith et al. [39] and Spiteri and Soler [40] focus on disaggregating 3 components: reformulation of existing food products by manufacturers, the net effect of the launching/removal of products, and consumer switching between products [39,40]. Because our study is focused on the changes in total beverage intake after the policy, we attempt to disaggregate only the effects of consumer behavior change from the total effects of reformulation. Our study provides a straightforward translation of results in grams of sugar or kilocalories per capita per day.

Our results indicate an increase in sugar, energy, and volume intake of untaxed beverages after the tax directionally consistent with but greater in magnitude than increases in a study in Berkeley that measured changes in both taxed and untaxed beverage consumption [21]. Water was a major component, accounting for 177 ml/capita/day (52%) of the increase in untaxed beverage volume (S3 Table). Part of this effect could be seen as a shift away from taxed to

untaxed beverages, as found in other studies [19–21,34]. However, Cape Town, South Africa, experienced a drought and severe water use restrictions from March to September 2018 [41]. Therefore, we cannot disentangle the tax-related effects from the effects of the drought on water consumption.

Analyzing results by socioeconomic status, sugar, energy, and volume intake decreased from taxed beverages and increased from untaxed beverages for all groups (S5 Table). However, there were no differences in absolute changes between groups, likely because the sample as a whole is relatively low income. For comparison, a recent pre–post study using purchase data from a nationally representative South Africa sample found greater reductions in sugar consumption (−32.7%) in LSM 4–6—nearly the identical range to our sample—compared to the higher socioeconomic group of LSM 7–10 (−20.4%) [42].

This study has several limitations. Given our data are cross-sectional, we are not able to follow all individuals over time, only measure differences in population means. Social desirability bias could have affected reporting and caused us to underestimate SSB intake. It is also possible that after the tax, social norms may have shifted so that the effect of social desirability bias is even greater after SSBs are subject to tax, causing an overestimation of reductions in SSB intake in this population. The magnitude of changes in taxed beverage consumption after the HPL in this study may not be generalizable to higher income populations, given that other studies have found the largest changes in sugary beverage intake among the lowest income groups analyzed [34]. However, the methods used in this study are useful for future work as they can separate the 2 crucial components of behavioral change and reformulation, regardless of the total effect of a tax on income groups. Finally, although we examined pre–post differences in beverage intakes under 2 scenarios, first accounting for differences in intake only, followed by accounting for differences in intake plus reformulation effects, we cannot isolate the specific types of reformulation that may have occurred, nor identify any causal effects. For example, changes may be due to reduced sugar content within taxation categories, or products may have switched categorization from taxed to untaxed if they were reformulated below the 4 g/100 ml threshold. To the extent that we did not carefully capture individual beverage heterogeneity within a wider brand-product-specific categorization weighting approach using Kantar data, we might misrepresent some of the nuanced industry changes.

Our study has several strengths, including a large sample of high-consuming young adults to increase study power and the ability to detect changes; the use of dietary intake data, which are a more suitable measure for mean population intakes than frequency questionnaires [33]; the use of a 2-part model for beverage intake [43]; and the development of time-varying FCTs linked with these dietary data to estimate changes in sugar, energy, and volume intake after the HPL.

## Conclusion

Using a large sample of a high-consuming, low-income population, we found large reductions in taxed beverage intake, separating the components of behavioral change from reformulation. This reduction was partially compensated by an increase in sugar and energy intake from untaxed beverages. Because policies such as taxes can incentivize reformulation, our use of time-specific beverage FCTs that reflect a rapidly changing food supply is novel and important for evaluating future taxation policies' impact on dietary intake.

## Supporting information

**S1 STROBE Checklist.**
(DOC)

**S1 Protocol. Study protocol.**
(DOCX)

**S1 Survey. Diet survey.**
(PDF)

**S1 Table. Beverage classification system.**
(DOCX)

**S2 Table. Model adjusted predicted intake of total sugar for taxed and untaxed beverage subcategories, Langa adults aged 18–39 years.**
(DOCX)

**S3 Table. Model adjusted predicted intake of energy (kilocalories) for taxed and untaxed beverage subcategories, Langa adults aged 18–39 years.**
(DOCX)

**S4 Table. Model adjusted predicted intake of volume for taxed and untaxed beverage subcategories, Langa adults aged 18–39 years.**
(DOCX)

**S5 Table. Model adjusted predicted intakes of total sugar, energy, and volume from taxed beverages for each LSM category.**
(DOCX)

## Acknowledgments

We wish to thank Karen Ritter for exceptional assistance with the data management, and Jessica Ostrowski and Bridget Hollingsworth for research assistance, Safura Abdool Karim for legal review of the legislation and regulation, and Emily Yoon for administrative assistance. We want to thank Chuma Maqina, Kelly Leman Scott, Sarona Munyai, Sikhumbule Joni, Lamla Sajini, Sisa Honesi Martins, Olwethu Mateta, Siyabulela Mditshwa, Andiswa Magwaza, Abongwe Benge, Ayakha Sigwabe, Masingita Nyalungu, Namhla Baskiti, Andisiwe Siramza, Siphesihle Thotsho, Nomhle Poni, and Ntombizodwa Nako for dietary intake and anthropometry collection, as well as Mawande Nelani, Pamela Magubane, Emmanuel Obasa, Zukiswa Mfinyongo, Ziyanda Lufele, Zikhona Lobola, Asisipho Majongile, Nqabakazi Majeke, Yanga Gadu, Busiswa Mqhayi, Priscilla Nkhensani, Thandokazi Magqaza, Onele Maqambayi, Nonelwa Ngcingane, Tembakazi Msutwana, Sisipho Mabaso, Chumani Matiwane, Akhona Rasmeni, and Sisipho Mngqnge for household questionnaires. We also want to thank Luchrechia Afrika, Vashika Chibba, Thandokazi Mahuzi, Kholiswa Manisi, Tyler Coats, Valentine Khumalo, Aneeqah Latief, Zintle Phekana, Stephanie Röhrs, Sharna Solomons, and Morongoa Tlhako, who conducted nutrition label data collection in 2018 and 2019. For the software used to collect and enter the nutrition label data, we thank the National Center for Advancing Translational Sciences (NIH UL1TR001111). We also thank The George Institute and Discovery Vitality for sharing early nutrition label data. For the use of the South African Food Data System FCTs, we thank the South African Medical Research Council.

## Author Contributions

**Conceptualization:** Michael Essman, Lindsey Smith Taillie, Shu Wen Ng, Barry M. Popkin, Elizabeth C. Swart.

**Data curation:** Tamryn Frank, Elizabeth C. Swart.

**Formal analysis:** Michael Essman.

**Funding acquisition:** Shu Wen Ng, Barry M. Popkin, Elizabeth C. Swart.

**Investigation:** Tamryn Frank, Shu Wen Ng, Barry M. Popkin, Elizabeth C. Swart.

**Methodology:** Michael Essman, Lindsey Smith Taillie, Shu Wen Ng, Barry M. Popkin, Elizabeth C. Swart.

**Project administration:** Tamryn Frank, Shu Wen Ng, Elizabeth C. Swart.

**Supervision:** Lindsey Smith Taillie, Shu Wen Ng, Barry M. Popkin, Elizabeth C. Swart.

**Writing – original draft:** Michael Essman.

**Writing – review & editing:** Michael Essman, Lindsey Smith Taillie, Shu Wen Ng, Barry M. Popkin, Elizabeth C. Swart.

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
