## [Editor Report · Decision Letter 0]

12 Aug 2020

Dear Dr Swart, 

Thank you for submitting your manuscript entitled "Taxed and untaxed beverage consumption by young adults in Langa, South Africa before and one year after a national sugar-sweetened beverage tax" for consideration by PLOS Medicine.

Your manuscript has now been evaluated by the PLOS Medicine editorial staff [as well as by an academic editor with relevant expertise] and I am writing to let you know that we would like to send your submission out for external peer review.

Kind regards,

Adya Misra, PhD,

Senior Editor

PLOS Medicine

---

## [Decision Letter · Decision Letter 1]

2 Nov 2020

Dear Dr. Swart,

Thank you very much for submitting your manuscript "Taxed and untaxed beverage consumption by young adults in Langa, South Africa before and one year after a national sugar-sweetened beverage tax" (PMEDICINE-D-20-03807R1) for consideration at PLOS Medicine. 

[LINK]

In light of these reviews, I am afraid that we will not be able to accept the manuscript for publication in the journal in its current form, but we would like to consider a revised version that addresses the reviewers' and editors' comments. Obviously we cannot make any decision about publication until we have seen the revised manuscript and your response, and we plan to seek re-review by one or more of the reviewers. 

We expect to receive your revised manuscript by Nov 23 2020 11:59PM. Please email us (plosmedicine@plos.org) if you have any questions or concerns.

We look forward to receiving your revised manuscript. 

Sincerely,

Adya Misra, PhD

Senior Editor 

PLOS Medicine

plosmedicine.org

Please revise your title according to PLOS Medicine's style. Your title must be nondeclarative and not a question. It should begin with main concept if possible. "Effect of" should be used only if causality can be inferred, i.e., for an RCT. Please place the study design ("A randomized controlled trial," "A retrospective study," "A modelling study," etc.) in the subtitle (ie, after a colon).

Abstract: could we perhaps rephrase “growing burdens of obesity and diabetes” I wonder if this is a bit stigmatising and could be revised 

Please clarify what you mean here “We estimated beverage intake using a two-part model, with a probit model for the first part and a generalized linear model with log-link for the second part”. Please also outline briefly how you planned to test behavioural versus reformulation effects 

The Data Availability Statement (DAS) requires revision. For each data source used in your study: 

Please use square brackets for references

Please add p-values of up to three decimal places, providing exact values throughout unless p<0.001

Please provide copies of questionnaires used or citations if they have been previously published

Did your study have a prospective protocol or analysis plan? Please state this (either way) early in the Methods section.

Table 1- I suggest you add the units for “Age”

Discussion- please start this section with a brief description of what was done and the main findings 

I suggest the discussion is tempered, by adding “our results indicate” rather than “we show” due to the observational nature of this study 

Please also temper all assertions of primacy by adding “ to our knowledge”

Please remove the funding information from the main text and add this information to the financial disclosure section to the article meta-data 

Please ensure that the study is reported according to the STROBE guideline, and include the completed STROBE checklist as Supporting Information. Please add the following statement, or similar, to the Methods: "This study is reported as per the Strengthening the Reporting of Observational Studies in Epidemiology (STROBE) guideline (S1 Checklist)."

Comments from the reviewers:

Reviewer #1: This is an excellent manuscript that provides a thorough evaluation of an important population health intervention. The methods that are reported are novel, and the focus on the low income group is important. This paper makes an excellent contribution to the literature.

I have some minor comments:

1. If the pre-tax data were collected two months before the implementation of the tax then some reformulation of soft drinks could have taken place already, which would result in an underestimate of the impact of the tax on sugar consumption. Our analysis of the impact of the UK sugar drink levy on sugar levels in drinks suggested that two months before the implementation of the levy the proportion of drinks with sugar levels over the 5g per 100mL levy threshold had already fallen by about 20 percentage points (Scarborough et al. PLoS Med, 2020, https://doi.org/10.1371/journal.pmed.1003025).

2. Lines 15—167: The description of how food diary data are linked with the nutrient composition tables is not very clear, and would be helped by an example. From my understanding, in the food diary data there was sometimes ambiguity at a brand-level (e.g. "Coca-Cola" could cover regular, cherry, mango, vanilla, etc.), and these multiple options were dealt with by weighting using Kantar World Panel data on consumption. If that is the case, then it would be useful to know whether this ambiguity extended to inclusion of both diet (i.e. zero calorie variants) and regular drinks amongst the same brand, which would be a big limitation.

3. The percentage split of men and women at each timepoint in table 1 are identical. Just checking whether this is a typo!

4. Lines 282-290: This should start "For total beverages" rather than "For untaxed beverages". However, the next bit of text does refer to 'untaxed beverages, so this paragraph needs a little bit of editing.

5. It is not clear to me how reformulation can have affected the volume of drink consumed. The only mechanism that I can think is that the respondents recorded how many units of drink they consumed (e.g. a bottle, a can) and the size of these units changed as a result of the tax. If so, then that should be made clear in the methods. Also, how were these units linked to specific serving sizes (e.g. in the UK, Coca-Cola is available in 250mL and 330mL cans. If a respondent reports consumption of 'a can of Coca-Cola' how do you choose which serving size to assign?)

6. I would like more information about how the nutrition data were collected. The manuscript simply says "First, nutrition facts panel (NFP) data were collected from South African grocery stores in February and March 2018. This was repeated exactly a year later in February and March 2019." How were the stores sampled? How were the products sampled? Was a complete audit conducted, or were only drinks that appeared in the food diary included?

Reviewer: Peter Scarborough

Reviewer #2: Thank you for the opportunity to review "Taxed and untaxed beverage consumption by young adults in Langa, South Africa before and one year after a national sugar-sweetened beverage tax." This manuscript uses a before and after study design to estimate the effect of the South African SSB tax on the consumption of sugar, energy and volume of both taxed and untaxed beverages, in a low-income adult community. The main strength of the paper is the use of detailed 24-hour dietary recalls linked with updated food composition databases for beverage. The paper is very clear, well structured and well written. 

In my read, the main limitation of the paper is the pre/post study design, as differences observed could be due to all sorts of time-varying factors, such as differences in the population sampled (either due to sampling process, change within the population, or change of the population, such as rising income), pre-existing decreasing trends in SSB purchases, changes in accessibility of beverages. In addition, the authors mention that there was a drought and severe water use restriction during part of the baseline data collection period. To account for some of these confounding effects, the authors adjust their results for a limited number of variables available, incl. age, day of the week, temperature, and a measure of socio-economic status, yet this does not rule out the presence of residual bias, such that the strength of the policy effect may remain uncertain. 

My specific comments and suggestions are as follows

1. Results summary in the abstract and conclusion: in my read, there seems to be a mismatch between what the paper says it intends to do and the results highlighted in the abstract. Both the introduction and the abstract indicate that the study aims to estimate changes in taxed and untaxed beverage intake. However, the only results presented in the abstract highlight the decrease in taxed beverage intake, showing that the tax might have had a positive effect on sugar and energy reduction. However, more than half of the reduction observed is compensated by what appears to be substitution towards untaxed beverages, which have dramatically increased in terms of overall volume consumption. This means that all in all, the SSB tax resulted in a relatively small reduction in sugar intake and an uncertain reduction in total energy from beverages in that population. 

2. I would remove the emphasis on the fact that looking at differences between reformulation and behavioural change is a novelty of this paper. Rachel Griffith et al. proposed a more sophisticated decomposition method in a 2014 IFS working paper "The importance of product reformulation versus consumer choice in improving diet quality". The paper makes the distinction between changes due to reformulation, behavioural change and introduction of new products. The method was applied to beverages by Marine Spiteri & Louis-Georges Soler in paper published in the European Journal of Clinical Nutrition in 2017 ("Food reformulation and nutritional quality of food consumption: an analysis based on households panel data in France"). 

3. Related the above point, how were new beverages introduced after the baseline treated in the analyses? 

4. It wasn't very clear to me how the nutritional values were attributed to beverages, in particular to what extent approximations had to be made using weights derived from purchase data. Could the authors provide more information on how detailed the beverage lists in the 24-hour recall was (i.e. how many beverage codes)? And how did the list of beverage codes in the 24-hour recall differ from the product level information available in Kantar? There is potentially some variability within beverage codes, which could affect the results. 

5. Could you indicate when the tax was announced in the text and in the timeline. In the UK, there was for example an announcement effect which triggered reformulation before the introduction of the tax. As a result, the total effect of the tax might not be fully captured by the analyses presented in the paper if reformulation started before the baseline data collection. This might deserve to be discussed. 

6. Could you explain a bit more on how the door to door sampling was conducted. Was there an element of randomisation for selecting the households to survey? Some important difference in socio-economic status were observed in the pre and post data collection waves. I was wondering whether the sampling method could have been responsible for part of those differences. You might have useful geographical location on the respondents at each wave, which could indicate systematic differences for example. Given the repeated cross-sectional study design, and the fact that few covariates are used in the models, I feel that I would need to be further reassured that the pre and post data collection do not systematically differ. 

7. I was wondering why other dietary information were not taken into account in the analyses. Could these for example provide an indication of the extent which the 2 samples systematically differ, which could then bias the results? Could these serve to adjust the results? 

8. Following the previous point, I was very curious to see a very large increase in the total volume consumed in the post-tax data collection in the adjusted model (+25%). How could this be explained? Was this also observed in the Kantar data analysed (paper under review cited)? Could this reflect a difference in socio-economic level between the 2 samples? Or rather the drought of 2018, as only briefly mentioned in the discussion. If the latter, this is potentially a massive confounding factor which should be further highlighted 

9. Table 1 - could you indicate what the LSM category mean in the table 

10. Why LSM was the only socioeconomic indicator used? Were other indicators available? 

11. It would be interesting to have an idea of the extent to which adjusted and unadjusted results differ. Maybe not necessarily in the main text.

12. Page 3 line 94: "while these data are important, they exclude many sources of SSBs which dietary measures overcome". Could you give an example of those sources of SSBs. 

13. What distributional assumptions are used for the second part of the model (for which a 'log-link' function is specified)? Do the data meet the assumptions? 

14. About the 2-part model: the type of model seems to make sense for the analysis of some food groups, which are rarely consumed. However, this might be less the case for total beverage volumes or sugar, or even reformulated products which contain a mixture of sugar and artificial sweeteners. I was wondering if the authors could reassure the reader that the results are not affected by this methodological choice, by reproducing some of the more general results using an alternative modelling strategy, where the 2 part-model does not feel as much justified. 

15. Could the authors provide all the coefficients of the fitted model for peer review purposes, and not just the adjusted predicted mean of the outcome. 

16. Sensitivity analysis. No change in energy intake when moving LSM from the analysis is no indication of an absence of bias, as both intake values could be biased indeed. To rule out whether results could be biased, the authors should assess whether missingness on LSM depends on the outcome (intake), conditional on the covariates (see e.g. Carpenter and Kenward, Multiple Imputation and its applications)

17. Page 8 line 325-326: could the authors avoid the word 'statistically significant', this only makes sense with reference to a certain level of significance. Also, the sentence seems to indicate that there was strong evidence of energy reduction overall, whereas the 95%CI includes 0, and the p-value (which is not reported) appears to be higher than 0.01. 

18. Page 5 line 170 "our key outcome variables include mean adjusted" - I am not sure what the authors mean by this. Outcomes (i.e. variables) are intake, not "mean adjusted" intake (i.e. summary statistic)? 

19. Figures 2-4. From the caption and figures it is not very clear what behavioural, and reformulation mean. I would be more explicit so that the figure can be read independently. 

20. Why didn't you run the analysis on households who were interviewed at each wave as a sensitivity analysis, as opposed to running it on those who were not. Focusing on the 'longitudinal sample' would ensure that the population is the same, which would rule out the concern that the populations are different between the 2 waves. 

Reviewer #3: This is a well-conducted study on beverage consumptions by young adults in Langa, South Africa before and after a national sugar-sweetened beverage tax. The study design, datasets, statistical methods and analyses, and presentation of results (tables and figures) are mostly adequate. However, there are a few issues especially interpretation of results needing attention.

1) The title and the abstract. The title is "Taxed and untaxed beverage consumption by...", however in the abstract, it's all about taxed beverage consumption in the findings and conclusion and nothing about untaxed and total beverage consumption. It's an incomplete summary/finding which doesn't match the title.

2) Interpretation of study results. Table 2 is a key table summarising the main findings of the study. We can see clearly that after introducing the SSB tax there were decreases in sugar, energy and volume consumption in taxed beverage but increases in untaxed beverage, and overall slight or no change in total beverage consumption. So far the paper is mainly focused on the results of taxed beverage consumption changes but largely ignored the untaxed and total beverage results, which is inadequate. To evaluate the effect of SSB tax fairly a holistic approach of presenting and interpreting all the results including taxed, untaxed and total beverage consumptions is needed in order to provide a complete picture of the effectiveness and impact of SSB tax policy. Balanced presentation and interpretation of results are needed in discussion, conclusion and in abstract.

3) The two-part model is key in estimation of beverage intake. Authors said the reason why the model was used is in the first part to account for beverage groups that have a high percentage of non-consumers (likelihood of consumption). Can authors make it explicit on what 'non-consumers' means? Were these young people consumer or non-consumer? What's the percentage and how it was modelled? Unless all these are made clear, readers will have doubts on the validity of using this two-part model.

[LINK]

---

## [Decision Letter · Decision Letter 2]

4 Feb 2021

Dear Dr. Swart,

Thank you very much for re-submitting your manuscript "Changes in taxed and untaxed beverage intake by South African young adults after a national sugar-sweetened beverage tax: a before-and-after study" (PMEDICINE-D-20-03807R2) for review by PLOS Medicine.

I have discussed the paper with my colleagues and the academic editor and it was also seen again by reviewers below. I am pleased to say that provided the remaining editorial and production issues are dealt with we are planning to accept the paper for publication in the journal.

Editorial points:

Thank you for addressing the previously stated issues. Before we proceed, please address the following issues:

• Please revise your title according to PLOS Medicine's style. Your title must be nondeclarative. For example, “Taxed and untaxed beverage intake by South African young adults after a national sugar-sweetened beverage tax: a before-and-after study.”

• Under the Competing Interests statement, please declare that Barry Popkin is an Academic Editor with PLOS Med.

• In the Author Summary, please avoid assertions of primacy. Specifically, the second bullet point under “Why was this study done?”, use the phrase, “To our knowledge, this is the first…”

• Your study is observational and therefore causality cannot be inferred. Please remove language that implies causality throughout the manuscript, such as “The majority of this increase was due to ...” Refer to associations instead. For example, in the third and forth bullet point under “What did the researchers do and find?”, line 359, line 378 to 390, line 393, Discussion section, and so forth.

• Please attenuate you statement in line 539 to 540.

• Change Methods, line 3 to rename Appendix A as Supporting Information file A or 1 and so forth.

• In Table 2 and 5, please provide exact p-values unless p <0.001

• In the tables, please explain, in the legend, hat the error bars represent

• Do not underline the text in line 445-446

• Please add the following statement, or similar, to the Methods: "This study is reported as per the Strengthening the Reporting of Observational Studies in Epidemiology (STROBE) guideline (S1 Checklist)."

• Under references, please indicate the access date for all websites consistently.

• Please update reference 42 – it is no longer under review.

• Indicate that Supporting Information file “Diet final” is a survey. Consider renaming it as such.

• In the third bullet point of 'What did the researchers find?' there is a typo, where 'non-taxed beverages' are referred to as 'taxed beverages.’ Please revise this.

[LINK]

We look forward to receiving the revised manuscript by Feb 11 2021 11:59PM.   

Sincerely,

Beryne 

PLOS | OPEN FOR DISCOVERY

Beryne Odeny MD, MPH, PhD(c) 

Associate Editor | PLOS Medicine

1265 Battery Street, Suite 200, San Francisco, CA 94111

bodeny@plos.org

plos.org | Facebook | Twitter | Blog

Requests from Editors:

Comments from Reviewers:

Reviewer #1: The authors have fully addressed all of my concerns at the first review round and I have only one very small further comment to make. In the third bullet point of 'What did the researchers find?' there is a typo, where 'non-taxed beverages' are referred to as 'taxed beverages'.

Thank you for the opportunity to review this paper.

Peter Scarborough

Reviewer #2: It was a pleasure to read the revised version of the manuscript. The authors have done an excellent job in systematically addressing all my initial comments and concerns, and I have no further comments. 

The paper looks stronger in my view and will be a nice contribution to the journal. 

Reviewer #3: Thanks authors for their great effort to improve the manuscript. I am satisfied with the response and revision. No further issues needing attention.

[LINK]

---

## [Editor Report · Decision Letter 3]

24 Feb 2021

Dear Dr Swart, 

On behalf of my colleagues and the Academic Editor, Dr. Sanjay Basu, I am pleased to inform you that we have agreed to publish your manuscript "Taxed and untaxed beverage intake by South African young adults after a national sugar-sweetened beverage tax: a before-and-after study" (PMEDICINE-D-20-03807R3) in PLOS Medicine.

PRESS

Sincerely, 

Beryne Odeny

Associate Editor 

PLOS Medicine